# Folate and Cobalamin Deficiencies during Pregnancy Disrupt the Glucocorticoid Response in Hypothalamus through *N*-Homocysteinilation of the Glucocorticoid Receptor

**DOI:** 10.3390/ijms24129847

**Published:** 2023-06-07

**Authors:** Arnaud Michel, Tunay Kokten, Lynda Saber-Cherif, Rémy Umoret, Jean-Marc Alberto, Déborah Helle, Amélia Julien, Jean-Luc Daval, Jean-Louis Guéant, Carine Bossenmeyer-Pourié, Grégory Pourié

**Affiliations:** 1Inserm UMRS 1256 NGERE-Nutrition, Genetics, and Environmental Risk Exposure, University of Lorraine, F-54000 Nancy, France; arnaud.michel@univ-lorraine.fr (A.M.); tunay.kokten@univ-lorraine.fr (T.K.); lynda.saber-cherif@univ-reims.fr (L.S.-C.); remy.umoret@univ-lorraine.fr (R.U.); jean-marc.alberto@univ-lorraine.fr (J.-M.A.); deborah.helle@univ-lorraine.fr (D.H.); amelia.julien@univ-lorraine.fr (A.J.); jean-luc.daval@univ-lorraine.fr (J.-L.D.); jean-louis.gueant@univ-lorraine.fr (J.-L.G.); gregory.pourie@univ-lorraine.fr (G.P.); 2National Center of Inborn Errors of Metabolism, University Regional Hospital of Nancy, F-54000 Nancy, France

**Keywords:** one-carbon metabolism, hypothalamus, fetal programming, post-translational modification, corticoid pathway

## Abstract

Vitamin B9 (folate)/B12 (cobalamin) deficiency is known to induce brain structural and/or functional retardations. In many countries, folate supplementation, targeting the most severe outcomes such as neural tube defects, is discontinued after the first trimester. However, adverse effects may occur after birth because of some mild misregulations. Various hormonal receptors were shown to be deregulated in brain tissue under these conditions. The glucocorticoid receptor (GR) is particularly sensitive to epigenetic regulation and post-translational modifications. In a mother–offspring rat model of vitamin B9/B12 deficiency, we investigated whether a prolonged folate supplementation could restore the GR signaling in the hypothalamus. Our data showed that a deficiency of folate and vitamin B12 during the in-utero and early postnatal periods was associated with reduced GR expression in the hypothalamus. We also described for the first time a novel post-translational modification of GR that impaired ligand binding and GR activation, leading to decrease expression of one of the GR targets in the hypothalamus, AgRP. Moreover, this brain-impaired GR signaling pathway was associated with behavioral perturbations during offspring growth. Importantly, perinatal and postnatal supplementation with folic acid helped restore GR mRNA levels and activity in hypothalamus cells and improved behavioral deficits.

## 1. Introduction

Nutritional conditions highly impact brain development. Current evidence supports that maternal hypovitaminosis of folate (vitamin B9) and cobalamin (Cbl, vitamin B12) predispose offspring to a wide range of metabolic and neurological disorders [1,2,3]. Maternal folate and/or Cbl deficiency induces an accumulation of homocysteine in plasma and neuronal cells and a decreased S-adenosylmethionine/S-adenosylhomocysteine (SAM/SAH) ratio, markers particularly associated with epigenetic mechanisms, neurotoxic effects, and brain diseases [4]. Various mechanisms have been proposed to explain observed deleterious effects, such as imbalanced proliferation and differentiation of neurons and brain atrophy leading to cognitive decline and a durable behavioral change in offspring [5,6,7,8,9].

In the context of maternal deficiency in methyl donors, fetal programming is a mechanism described in specific brain function defects, such as locomotion coordination, learning, and memory [10,11]. Nevertheless, numerous pathways are involved after the deregulation of folate and/or methionine cycles [12,13,14,15,16]. Concerning brain outcomes, conclusions sometimes need to be completed because brain circuits are interconnected, and a function typically attributed to a specific circuit could be influenced by other circuits, especially in early life in the case of avitaminosis [17].

The hypothalamus plays a central role in regulating peripheric functions but also other central circuits using endocrine signals. Indeed, besides food intake regulation [18], the hypothalamus signals modulate stress response, depressive-like behaviors, and, more largely, cognitive outcomes [19,20,21]. Moreover, developmental studies highlighted the fetal programming phenomenon in the predictive status of offspring concerning hypothalamic-related functions [22,23,24]. As in other brain sub-areas, hypothalamus circuits develop and connect synapses during the embryonic and postnatal periods, two sensitive time windows for hormonal and environmental influences putatively leading to fetal programming consequences [25,26]. Using a mother–offspring rat model of gestational methyl donor deficiency (MDD), we previously showed that a reduced availability in methylation status led to body growth retardation [9,26], abnormal hypothalamic development, and impaired neuropeptides expression in the arcuate nucleus (ARC) and the ventromedial nuclei (VMH) [27].

Nevertheless, a perinatal folate supplementation restoring the methylation status could alleviate such physiopathologic effects [9,27]. Besides regulating food intake, the hypothalamus is implicated in stress response through the hypothalamic–pituitary–adrenal (HPA) axis leading to physiological and behavioral adjustments, at least in part with glucocorticoids [28,29]. This hormonal regulation also produces a feedback mechanism to the hypothalamus. The glucocorticoid receptor (GR) is a major factor in many cellular regulations and gene expressions with other chaperone proteins, such as FKBP51 and HSP90 [30,31]. Glucocorticoids (GCs) are also key hormones regulating food intake and energy balance through neuropeptides NPY/AgRP [32,33]. Conversely, GCs inhibit the release of corticotropin-releasing hormone (CRH), an anorexigenic neuropeptide decreasing the NPY/AgRP synthesis and thus decreasing appetite [34,35]. As for other mediators, it has been shown that a down-regulation of the one-carbon metabolism is associated with epigenomic modulation of the GR pathways, a decreased interaction with ligands in the hippocampus, and cognitive dysfunction [36].

Thus, considering that the hypothalamus is implicated in numerous neuronal functions, that (i) the formation of its circuits occurs in the key time window around birth, (ii) GCs play a central role in gene expressions in hypothalamic neurons, it appears relevant to investigate if the GR mediation could modulate the cellular pathways in the ARC and VMH hypothalamic regions in the context of a pre and postnatal methyl donor deficiency. In the present study, we explored the impact of a maternal methyl donor deficiency (MDD) during gestation and lactation on the expression and activation of GR in both ARC and VMH hypothalamic regions of offspring. In addition, we investigated whether a perinatal supplementation with folic acid can modify the GR pathway in deprived rats and its influence on neuropeptide expression.

## 2. Results

### 2.1. Effect of Vitamin B9 on One-Carbon Metabolism Markers and Expression of Glucocorticoid Receptor (GR) in the Hypothalamus

As previously documented, maternal nutritional methyl donor deficiency during gestation and lactation affects plasma levels of folate and vitamin B12, which are dramatically reduced in the rat progeny at weaning (postnatal day 21). In parallel, homocysteinemia is significantly augmented (*p* < 0.01). Table 1 shows that folic acid supplementation restores folate concentration without affecting vitamin B12 status and significantly reduces hyperhomocysteinemia in deficient pups.

Investigating the expression of mRNA and protein levels of GR, our results show that a deficiency in methyl donor (vitamin B9 and/or B12) leads to significantly reduced mRNA by 50% and protein by 32% of this receptor (Figure 1 and Figure 2A) in the hypothalamic cell line and hypothalamus brain tissue. Supplementing rat offspring with folic acid (FA) surprisingly led to a significant reduction of mRNA by 24% in the hypothalamus of controls. Still, such an FA-supplementation partially restores GR mRNA in MDD offspring by 29% compared with not-supplemented MDD (Figure 2A). In addition, the microscope observation of hypothalamic tissues shows a cellular delocalization of GR protein between offspring groups. Indeed, young MDD rats present a cytoplasmic localization of GR in the ARC and VMH compared with controls. Cytoplasmic GR aggregates are also observed in MDD rats. FA-supplementation led to relocalizing GR in nuclei (Figure 2B).

### 2.2. B9-Deficiency Leads to N-Homocysteinilation of GR and Loss of Function

We analyzed the peptides collected after immunoprecipitation with anti-Hcy antibody from the rat forebrain by mass spectrometry. The N-homocysteinylation of one lysine residue (K699) of the 686–699 peptide and one threonine phosphorylation at residue T686 are identified (Figure 3A). Both modifications are located in the ligand-binding domain of GR. This domain is highly conserved, and the 686–699 rat peptide has a 100% sequence identity with mouse GR (683–696) and 93% sequence identity with human GR. A glutamic acid (E696 rat/E693 mouse) was replaced by an aspartic acid in the human sequence (D 678) (Figure 3).

A prominent interaction of homocysteine with GR was confirmed by the Duolink in situ Proximity Ligation Assay that shows a significant increase of fluorescent signals in deficient hypothalamic cells compared with controls (Figure 4A). Because the procedure is based on a stoichiometric reaction, each red dot corresponds to a close interaction between GR and homocysteine. Furthermore, the functional adverse effects associated with increased GR homocysteinylation in the ligand-binding site are reflected by a 24% reduced expression of the GR response gene coding for AgRP (Figure 4B).

The prominent interaction of homocysteine with GR was confirmed by the Duolink in situ Proximity Ligation Assay in our rat model. GR N-homocysteinylation is significantly increased in the arcuate nucleus of deficient rats (MDD) compared with controls (152% increase). B9-supplementation significantly reduces the GR N-homocysteinylation in MDD rats (35% reduction) but not at the control levels (Figure 5A). Furthermore, we analyzed the dimerization and the nuclear localization of GR. GR–GR interaction is significantly reduced in the arcuate nucleus of deficient rats (34% in the cytoplasm and 66% in nuclei; Figure 5B). Moreover, this was associated with a 25% reduced GR response gene coding expression for AgRP. The AgRP mRNA level is significantly restored by the B9-supplementation in MDD rats (Figure 5C).

### 2.3. Behavioral Characterization of Rat Offspring

The general activity was quantified during the suckling period in three-time points after the first environmental explorations that offspring presented from the nest. Thus, rat offspring usually increase their locomotion during the second half of the suckling period, which was confirmed in our results for the control group from D11 to D18. In contrast, MDD young rats showed a lower number of zones visited, and such a low activity remained equal at all tested ages. FA-supplementation did not modify control activity, but the number of zones visited at D18 by MDD-FA offspring reached the control groups (Figure 6A).

Olfaction sense usually follows the same profile as general activity. By testing the offspring’s abilities to recognize a familiar odor, our results showed that control rats increased the time spent in the familiar odor zone while MDD did not, whatever the age. FA-supplementation did not modify control results but led to a hugely significant increase in the treated MDD group at D18 (Figure 6B). The same profile of result was obtained by quantifying the olfactory success. Indeed, control offspring obtain the best percentage of success in recognizing a familiar odor compared with MDD. FA-supplementation tends to improve the control and MDD success scores at D18 compared with their respective untreated groups (*p* = 0.0620 and 0.0757, respectively). An interesting measurement of the olfactory improvement curve from D11 to D18 (with a linear formula) showed that all groups display a positive progression between +11 to +25% slope, except the MDD group showing a −11% decreasing slope. Thus, FA-supplementation switched the olfaction performance of MDD from negative to positive while offspring developed (Figure 6C).

## 3. Discussion

A deficiency in methyl donors, such as vitamin B9 (folate) and/or B12, especially during pregnancy, is related to neurodevelopmental defects in offspring. For several decades, this led to a supplementation program for pregnant women in some countries or food fortification in others [38,39,40]. Considering the brain specificity, which still develops and maturates its circuits after birth, several animal studies have also suggested a prolongation of vitamin status correction to favor a better birth status [41] or brain plasticity and maturation after birth [9,11]. Thus, some brain sub-regions still complete their circuits formation and connectivity based on a so-called Hebbian mechanism, even long after birth, while young individuals are subjected to internal and environmental factors [11,42]. The present study deals with such mechanisms, including environmental and biochemical factors that could influence the development and maturation of some brain sub-structure even after birth. Nevertheless, as brain sub-structures could be interconnected, the functions of one could influence others.

The hypothalamus is a neurovegetative center that receives numerous external and internal inputs and regulates brain and other organs’ functions through neuroendocrine messages [43]. A direct hypothalamus influence on cognitive brain areas and functions such as memory and anxiety has been recently shown [44]. Due to its major central influence, the hypothalamus could be implicated in deregulations of brain formation and/or functions under hyperhomocysteinemia.

Several of our previous studies have shown that methyl donor deficiency affects different brain functions and behaviors, such as olfaction performances [45], early locomotion and coordination [10], and hippocampus-related cognition [46]. Moreover, it has been shown that specific receptors could be linked to neuronal mechanisms and methyl donor deficiency, such as Estrogen receptor alpha [10] or glucocorticoid receptor [47]. Because the hypothalamus uses the glucocorticoid receptor and plays its influences other brain circuits, modulation of the glucocorticoid pathway could lead to a cascade of influences between circuits linked to the hypothalamus.

The results of the present study have shown that a methyl donor deficiency during the perinatal period led to a decrease of GR mRNA and protein in hypothalamic cells and that an FA-supplementation reversed this effect. As an additional consequence, a lack of nucleus translocation of GR was quantified in deficient rats. Additionally, deregulation of the one-carbon metabolism increased the circulating and tissue homocysteine, a metabolite identified as a major cytotoxic marker [4]. One of the related cellular deleterious mechanisms has been described as N-homocysteinylation of proteins leading to disturbing its function [48,49]. Our results showed that GR appears sensitive to N-homocysteinylation on a lysine residue in position 699, corresponding to the ligand-binding domain of the protein. We also quantified an elevation of GR–homocysteine interaction in hypothalamus cells submitted to a methyl-deficient status. Investigating the functional outcomes of the GR protein, our results showed that GR–GR dimerization and the translocation from cytoplasm to nucleus were reduced in deficient conditions and restored at least partially by an FA-supplementation. Using one of the target genes of GR in the hypothalamus, we finally showed that the deregulation of the GR pathway induced the decrease of AgRP mRNA in deficient rats, also restored by an FA-supplementation.

From a behavioral point of view, the implication of the hypothalamus has been described in food intake but also in cognitive functions [23]. Indeed, besides the well-known diet regulation, the diversity of hypothalamus circuit connections also modulates psychiatric-related outcomes, such as stress, anxiety, and depression syndromes [50]. A reduced olfaction capacity appears as a sense and behavioral marker linked to hypothalamic dysfunction and stress and anxiety syndromes. Indeed, the olfactory system projects neuronal connections to highly relevant brain sub-structures, such as the limbic system (i.e., amygdala and hippocampus) for cognitive functions and the thalamus and hypothalamus for vegetative and neuroendocrine regulations [51]. In the last decades, it has been shown that many psychiatric syndromes, such as stress, anxiety, and depression, could be related and predicted through olfactory cues or putative deficits or deregulations [52,53,54]. The behavioral investigations of the present study measured that general activity and olfaction performances were reduced in young rats submitted to B9/B12 deficiency during the perinatal period and that a late FA-supplementation partially restored such olfactory–hypothalamic-related behaviors.

Taken together, the results of the present study highlighted the deregulation of the glucocorticoid receptor in the hypothalamus for the first time by an N-homcysteinylation mechanism under a methyl donor deficiency status. GR was already shown to be implicated in hippocampus deregulation for learning and memory function in a B12-deficient model. Thus besides epigenetic mechanisms on the GR gene [55,56] and deregulations of proteins related to the GR pathway [36], the N-homocysteinylation of GR appears as an additional mechanism leading to deleterious neuronal effects in brain tissues in the context of hyperhomocysteinemia and methyl donor deficiency. This is especially relevant because the hypothalamus plays a central role in the maturation and functions of other brain sub-structures and influences the maturation and behavioral outcomes of many neuronal functions after birth. In addition, a prolonged FA-supplementation after the third trimester of gestation could rescue some deleterious effects, considering the knowledge of post-developmental specific windows of time after birth for brain maturation and post-formation [11,57].

## 4. Materials and Methods

### 4.1. Animals and Tissue Collection

In vivo experiments were performed on a validated animal model of methyl donor deficiency [5]. They were conducted according to the international guidelines for the care and use of laboratory animals and were approved by the local University Research Ethics Board (authorization number APAFIS#5509-2016053112249550). Wistar rats (Charles River, L’Arbresle, France) were maintained under standard laboratory conditions, on a 12 h light/dark cycle, with food and water available ad libitum. One month before mating, adult females were fed either a standard diet (Maintenance diet M20, Scientific Animal Food, and Engineering, Villemoisson-sur-Orge, France) or a methyl donor-deficient (MDD) low-choline diet (119 vs. 1780 mg/kg) lacking folate and vitamin B12 (Special Diet Service, Saint-Gratien, France). Methionine content (~0.4%) was similar in both diets. The supplementation protocol was conducted as previously described [26]. Briefly, folic acid (FA, Sigma-Aldrich, Saint-Quentin Fallavier, France) was diluted in condensed milk and given per OS at 3 mg/kg per day in a final volume of 1 mL to dams from embryonic days (E) 13 to 20. Matched control dams received the same volume of vehicle (i.e., 1 mL condensed milk) over the same period.

Offspring were kept until 21 days of age (weaning time) for behavioral investigations; then, they were euthanized by an excess of isoflurane. Brains were rapidly collected and laid on ice to carefully separate the two hemispheres according to the sagittal plane. Left hemispheres were dropped in cold (+4 °C) methylbutane to (i) remove physiological fluids, (ii) dehydrate, and (iii) freeze the tissues at −20 °C. Left hemispheres were finally stored at −80 °C for further cryo-sections. Right hemispheres were microdissected on ice, and brain sub-areas such as the hypothalamus were immediately frozen in liquid nitrogen and stored at −80 °C until biochemical/molecular analyses. For immunohistochemistry, twelve µm sagittal cryo-sections were generated from the left hemispheres, starting from the zero planes that bisect the brain mid-sagittally. Brain structures were identified according to the Paxinos and Watson atlas for slide standardization according to the same coordinates. For subsequent staining, selected slides were coded prior to analysis, and the codes were not broken until experiments were completed (blind procedure). Selected tissue sections were fixed in 4% paraformaldehyde 1 h at room temperature prior to the immuno-technique (see below).

### 4.2. Cell Culture

rHypoE11 (rat) neuronal cell lines purchased from American Type Culture Collection (ATCC) was conditionally immortalized by transfer of a temperature-sensitive simian virus 40 large tumors (SV40 T) antigen to primary hypothalamic neuronal cell cultures obtained from fetal mice on embryonic days E15, E17, and E18, and from E18 fetal rats [28]. Cells were cultivated in 1× Dulbecco’s modified Eagle’s medium (DMEM, D5796, Sigma-Aldrich, Saint-Quentin Fallavier, France) with 10% fetal bovine serum (CVFSVF0001, lot: S52751-2262, Eurobio, Courtaboeuf, France), 1% penicillin/streptomycin (P4333, Sigma-Aldrich), and maintained at 37 °C with 5% CO_2_. For homogeneity, the same serum was used throughout all experiments. The cells grew to form a monolayer culture attached to the culture plate and were split when they reached 70–90% confluence, with a plate ratio of 1:5. Because standard DMEM does not contain vitamin B12, methyl donor deficiency was induced by using a poor medium (DMEM D2429, Sigma-Aldrich) lacking vitamin B9 (folic acid), with the addition of 2 mM glutamine (G7513, Sigma-Aldrich), 3.7% sodium bicarbonate (S8761, Sigma-Aldrich), 0.35% glucose (G8769, Sigma-Aldrich), 10% fetal bovine serum and 1% penicillin/streptomycin. Cells were kept in B9-free conditions for 24 or 48 h before subsequent analyses.

### 4.3. Measurement of Maternal Plasma Concentrations of Homocysteine, Vitamin B12, and Folate and Offspring Tissue Concentrations of SAM and SAH

Homocysteine concentrations were measured by HPLC (Waters, St. Quentin, France) coupled with mass spectrometry (Api 4000 Qtrap; Applied Biosystems, Courtaboeuf, France). Vitamin B12 and folate concentrations were measured by radio-dilution isotope assay (simulTRAC-SNB; ICN Pharmaceuticals, Versailles, France) as previously described [58].

### 4.4. Immunohistochemistry

Immunohistological analyses were performed on brain sections at the level of the arcuate and the ventromedial nuclei of the hypothalamus according to the Paxinos and Watson rat brain atlas [59]. Nonspecific binding sites were blocked in phosphate-buffered saline containing 1% bovine serum albumin (BSA), and incubation was performed overnight with an antibody against the glucocorticoid receptor (mouse monoclonal, 1/200, Novus Biologicals, Centennial, CO, USA). After a washing step, immunoreactivity was assessed by incubation in the presence of an appropriate secondary anti-IgG antibody conjugated to AlexaFluor for 1 h at 25 °C (1/1000, Life Technologies, Saint-Aubin, France). Cell nuclei were counterstained with the DNA fluorochrome 4,6-diamidino-2-phenylindole (DAPI, Sigma-Aldrich). Control experiments were conducted by omitting the primary antibody. Immunofluorescence visualization, image acquisition (×20 and ×60 magnification), and unbiased cell counts in randomly selected fields were performed with a BX51WI microscope (Olympus, Rungis, France) coupled to a ProgRes MF cool camera (Jenoptik, Saint-Martin-des-Entrées, France) or a confocal microscope (Nikon Instruments, Champigny sur Marne, France) and analyzed by Cell^®^ software (version 3.1, Olympus, Rungis, France).

### 4.5. Western Blotting

Following protein extraction from brain extracts with RIPA buffer, Western blot analyses were performed using a standard procedure with chemiluminescence using the ECL system (Bio-Rad, Marnes-la-Coquette, France), as previously detailed [13]. Antibodies against the glucocorticoid receptor (mouse monoclonal, 1/1000, Novus Biological) and the Glyceraldehyde-3-phosphate dehydrogenase (GAPDH, chicken monoclonal, 1/1000, Millipore, Fontenay-sous-Bois, France) was used. GAPDH served as an internal standard. Polyvinylidenedifluoride membranes were incubated for 1 h at room temperature with the corresponding horseradish peroxidase-conjugated preadsorbed secondary antibody (1/2000, Santa Cruz, CA, USA).

### 4.6. RNA Extraction and Quantitative RT-PCR

RNA was purified from rat hypothalamus tissues. RNA extraction was performed with TRIzol^®^ (Invitrogen, Cergy-Pontoise, France) according to the manufacturer’s instructions. The concentration and purity of RNA were determined at 260/280 nm using a nanodrop spectrophotometer Multiskan GO (Thermo Fisher Scientific, Illkirch, France).

RNA (300 ng) was then subjected to a two-step RT-qPCR using the PrimeScript™ RT Master Mix and SYBR^®^ Premix Ex Taq^®^ (Takara, Ozyme, Saint-Cyr-l’Ecole, France) following the manufacturer’s instructions. Primers are detailed in Table 2 and were purchased from Eurogentec (Liège, Belgium). The amplification products were analyzed by agarose gel electrophoresis to confirm amplicon size and primer specificity (a single band at the expected size). Cycle threshold (Ct) was determined from each sample, and real-time PCR amplification efficiencies were expressed by calculating the ratio of crossing points of amplification curves. The expression of genes of interest was normalized to those of GAPDH/RPS29 for the rat species and Pol2/RPS29 for the mouse species using the 2−∆∆Ct method.

### 4.7. Mass Spectrometry

Coimmunoprecipitation experiments were performed with the control and rat forebrain using a Pierce^®^ Co-Immunoprecipitation kit (Thermo Fisher Scientific, Illkirch, France) to assess the Hcy interaction with proteins. Hcy coimmunoprecipitation was analyzed by mass spectrometry (MS), as described by Bossenmeyer-Pourié and colleagues (2019) [48]. Tryptic peptides were obtained from immunoprecipitated proteins and separated using a nano-chromatography system (Proxeon, Thermo Fisher Scientific) connected online to an LTQ Velos Orbitrap (Thermo Fisher Scientific) mass spectrometer. Separation conditions were as in Ren and colleagues [60]. MS spectra were acquired on the Orbitrap analyzer at resolution mode R = 30,000. After each MS spectrum, an automatic selection of the 20 most intense precursor ions was activated with a 15 s dynamic exclusion delay to acquire MS/MS spectra on the LTQ Velos analyzer using CID fragmentation mode at 35% relative resonant activation energy for 40 ms. Spectra were analyzed using Mascot 2.3.2 (Matrix Science). The search was performed against a non-redundant database of rodent protein sequences from Swissprot, to which corresponding random decoy entries were added. The mascot was run in MS/MS Ion search mode with the following parameter settings: no fixed modification, variable modifications (homocysteinylation on lysine and homocysteinylation on cysteine), precursor mass tolerance 10 ppm, fragment ions mass tolerance 0.6 Da, 2 missed cleavages, and trypsin as a digestion enzyme. Additional filtering was applied after protein and peptide identification for further analysis, using the following criteria: for single and multiple peptide hit proteins, each Mascot ion scores. Such criteria allowed a false-positive rate below 1% for each injection, based on the numbering of identified decoy entries.

### 4.8. Behavioral Evaluation

As part of a global neurofunctional approach, various behaviors, such as locomotion, exploration, and olfaction, were studied using a corridor apparatus. Because spontaneous exploration of rat offspring is significant after postnatal days 10, 11, 14, and 18 (D11, D14, and D18) were chosen. For this purpose, the maternal diets were as described above and maintained after delivery until the last testing day. As previously described, specific behavioral items revealing specific rat neuro-functional abilities were measured [45,61]. At each time point studied, rat pups performed one trial test of 300 s (cut-off) consisting in moving freely in a corridor apparatus (90 cm in length, walls of 15 cm high, and a corridor of 6 cm wide). As olfaction stimuli, 5 g of litter was randomly placed at each end of the corridor (fresh or from the home cage of individuals). Rats were prohibited from touching the litter samples using a wall containing holes. The number of rat pups in each group was as follows: 19 controls, 40 FA-controls, 14 MDD, and 33 FA-MDD. For homogeneity, tests were performed between 8:00 and 11:00 a.m. and video-recorded with a video-tracking system (Viewpoint, Lyon, France).

### 4.9. Statistical Analysis

Data were analyzed with Statview 5 software for Windows (SAS Institute, Berkley, CA, USA). Tissue data were compared using a one-way analysis of variance (ANOVA) with Fisher’s test. Behavioral data were compared using non-parametric tests because of the unequal numbers of animals in each group. Kruskal–Wallis or Mann–Whitney tests were applied according to the number of groups compared. A *p*-value < 0.05 was considered to indicate significance.

## Figures and Tables

**Figure 1 ijms-24-09847-f001:**
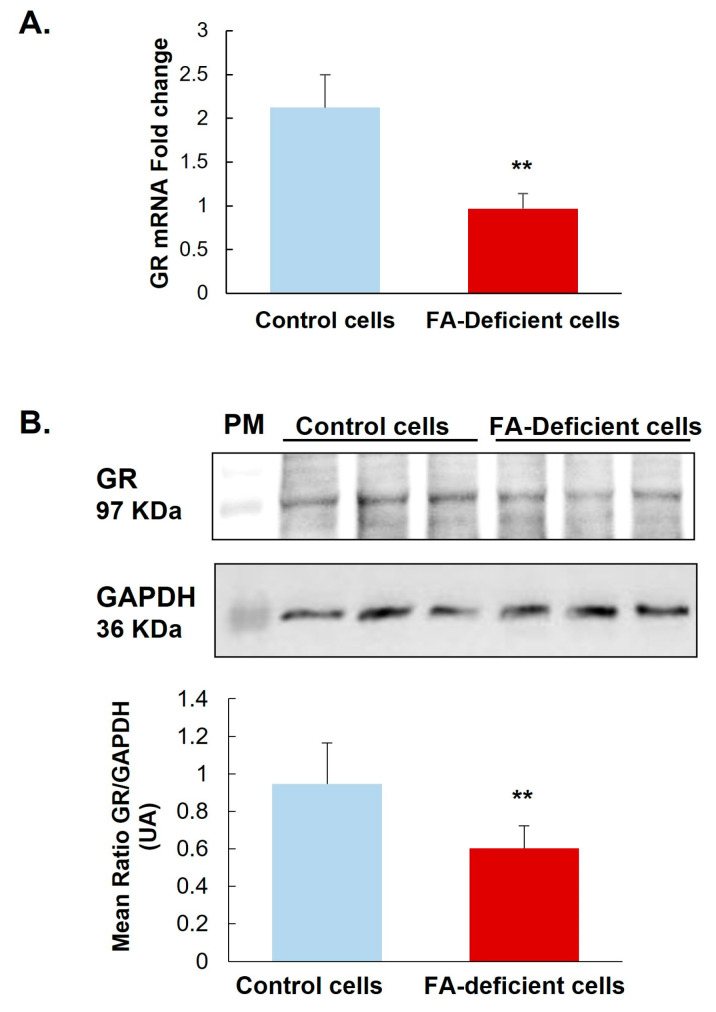
Effects of methyl donor deficiency on GR expression in a rat hypothalamic cell line. (**A**) GR mRNA expression 24 h after B9 deficiency. (**B**) GR protein expression 24 h after B9 deficiency. Data are means ± SD from 3 separate cultures and are reported as arbitrary units. Statistically significant differences between control and B9 deficient cells: ** *p* < 0.01.

**Figure 2 ijms-24-09847-f002:**
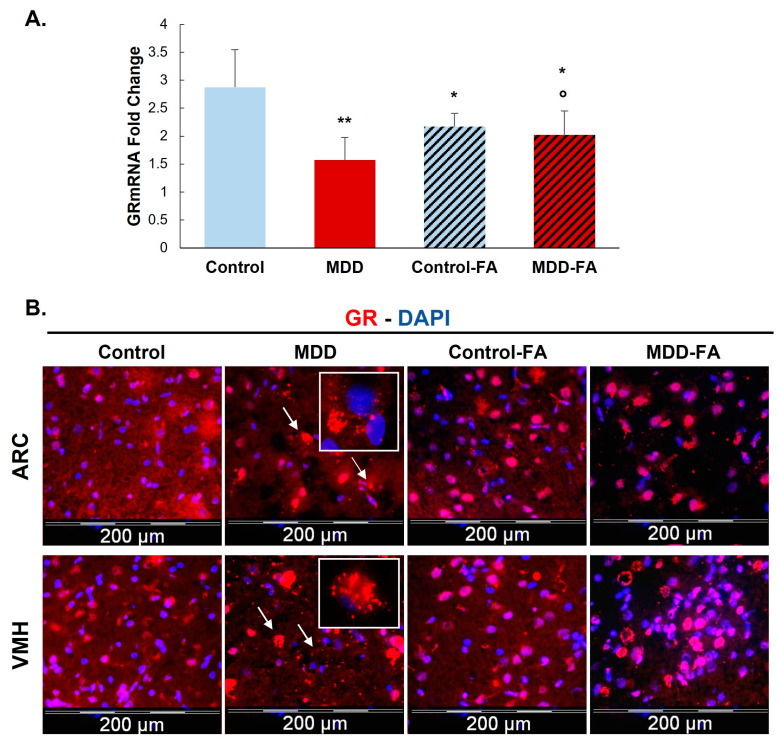
Effects of methyl donor deficiency and FA-supplementation on GR expression in the hypothalamus of rat pups at 21 days of age. (**A**) GR mRNA expression. Data are means ± SD from 6 ≤ *n* ≤ 8 animal samples and are reported as arbitrary units. Statistically significant differences between control and deficient (MDD) rats: * *p* < 0.05 and ** *p* < 0.01. Statistically significant differences between deficient (MDD) and FA-supplemented deficient (MDD-FA) rats: ° *p* < 0.05. (**B**) Immunohistochemistry of GR expression in the arcuate (ARC) and the ventromedial (VMH) nuclei of the hypothalamus of control and deficient (MDD) and FA-supplemented rat pups at 21 days. White arrows show aggregate-shaped GR proteins and white squares show a higher magnification (×100) of these aggregate-shaped GR proteins.

**Figure 3 ijms-24-09847-f003:**
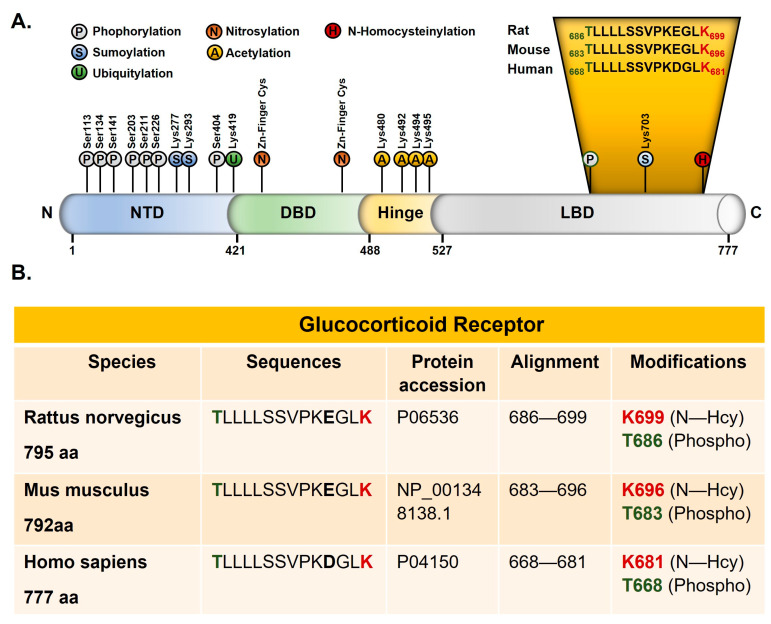
Post-translational modifications of GR. (**A**) New post-translational modification sites identified by mass spectrometry in the ligand-binding domain of the rat glucocorticoid receptor. N-homocysteinylated lysine is shown in red and phosphorylated threonine in green (drawn according our results and the review from Weikum et al. [37]). (**B**) Alignments of the rat, mouse, and human glucocorticoid receptor. NTD: amino-terminal domain, DBD: DNA-binding domain, and LBD: ligand-binding domain.

**Figure 4 ijms-24-09847-f004:**
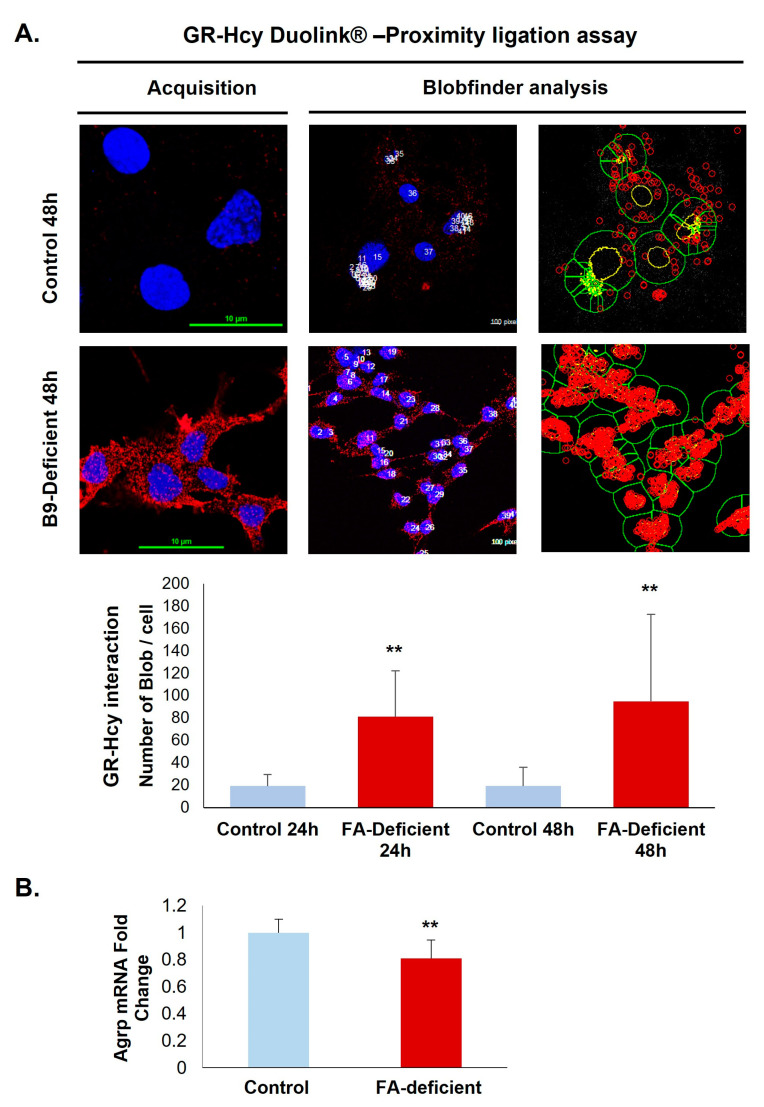
Quantification of the GR N-homocysteinylation in a rat hypothalamic cell line. (**A**) In situ interaction between homocysteine and GR was monitored by the Duolink assay (acquisition images) and the Blobfinder freeware (analysis example images) in control and vitamin B9-deficient cells. (**B**) AgRP mRNA expression analysis. All experiments were performed in triplicate. Data are reported as means ± SD (3 ≤ *n* ≤ 9). ** *p* < 0.01 vs. control.

**Figure 5 ijms-24-09847-f005:**
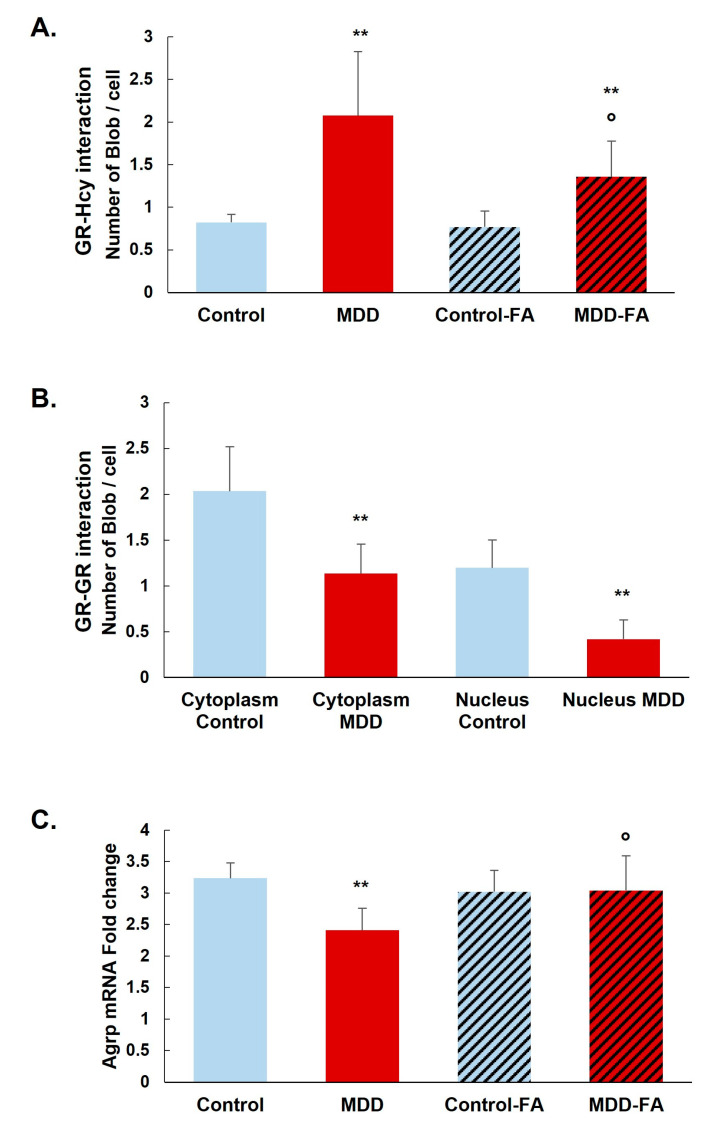
Effects of methyl donor deficiency and FA-supplementation on the quantification of GR N-homocysteinylation in the rat hypothalamus. (**A**) Quantification of GR N-homocysteinylation monitored by the Duolink assay and the Blobfinder freeware in the arcuate nucleus. (**B**) Quantification of GR–GR interaction monitored by the Duolink assay and the Blobfinder freeware in the arcuate nucleus. (**C**) Hypothalamic AgRP mRNA expression analysis. All experiments were performed in triplicate. Data are reported as means ± SD (3 ≤ *n* ≤ 9). Statistically significant differences between control and deficient (MDD) rats: ** *p* < 0.01. Statistically significant differences between deficient (MDD) and FA-supplemented deficient (MDD-FA) rats: ° *p* < 0.05.

**Figure 6 ijms-24-09847-f006:**
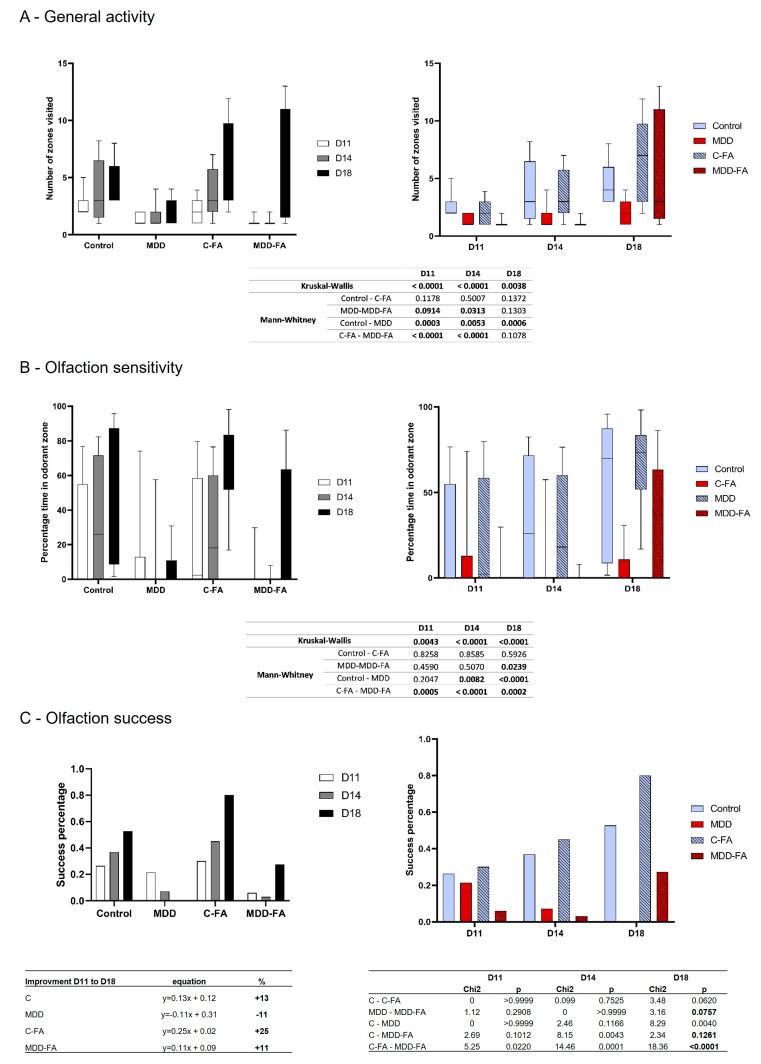
Effects of methyl donor deficiency and FA-supplementation on some markers of the 21-day-old pups’ behavior. Results are presented according to groups (black and white) or the age of growing pups (color). (**A**) Evaluation of the general activity level in terms of environmental exploration. (**B**) Evaluation of olfaction sensitivity. (**C**) Evaluation of olfaction recognition in terms of percentage of olfaction success. Data are reported as median and quartiles (14 ≤ *n* ≤ 40). Statistics: According to parameters, such as number of groups > 2, number of individuals in groups, and variability of data, a non-parametric analysis was performed (Kruskal–Wallis and Mann–Whitney tests); tables of statistical analysis are shown of each behavioral test.

**Table 1 ijms-24-09847-t001:** Effects of the maternal dietary regimen on plasma concentrations of folate, vitamin B12, and homocysteine in 21-day-old rat pups. Data are means ± SD, obtained from 15 ≤ *n* ≤ 40 individuals. Statistically significant differences: ** *p* < 0.01 with respective control and °° *p* < 0.01 between MDD and MDD + folic acid (MDD = methyl donor deficiency).

	Plasma Folate (nmol/L)	Plasma Vitamin B12 (pmol/L)	Plasma Homocysteine (µmol/L)
21d Control-Vehicle	75 ± 8.3	918.6 ± 54.7	5.8 ± 1.5
21d MDD-Vehicle	18.5 ± 6.2 **	389.3 ± 157.1 **	17.6 ± 6.3 **
21d Control + Folic Acid	111.17 ± 33.51 **	878.9 ± 141.16	4.7 ± 1.1
21d MDD + Folic Acid	98.1 ± 24.84 **/°°	397.2 ± 122.4 **	6.4 ± 2.1 **/°°

**Table 2 ijms-24-09847-t002:** Sequences of primers used for quantitative PCR.

Genes	Pimers	Sequences
AgRP	Forward	CGGAGGTGCTAGATCCACAGA
AgRP	Reverse	AGGACTCGT GCAGCC TTACAC
GR	Forward	TGAAGCTTCGGGATGCCATT
GR	Reverse	ATTGTGCTGTCCTTCCACTG
Pol II	Forward	AGCAAGCGGTTCCAGAGAAG
Pol II	Reverse	TCCCGAACACTGACATATCTCA
RPS 29	Forward	ATGGGTCACCAGCAGCTCTA
RPS 29	Reverse	CATGTTCAGCCCGTATTTGC

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
