# Peer review of "Folate and Cobalamin Deficiencies during Pregnancy Disrupt the Glucocorticoid Response in Hypothalamus through *N*-Homocysteinilation of the Glucocorticoid Receptor"

_ijms, 2023, doi:10.3390/ijms24129847_

Round 1
Reviewer 1 Report
Dear Authors,
Please find the comments in the attachment.
Sinceelry,
Reviewer

Author Response
Authors thank the reviewers for their help and advices improving the manuscript.
We apologize if the quality of English writing appears not satisfying. We use a software to improve our first composition for grammar and clarity. The revised manuscript has obtained 97% of clarity and quality of writing. In the revised version, all modifications appear highlighted for rapid comparison (pink for reviewer1 comments, green for reviewer2, and light blue for reviewer3).
Please find below, our responses to each reviewer's comment.
Reviewer1 round1
Manuscript title: Folate and vitamin B12 gestational deficiency disturb glucocorticoid response in hypothalamus through N-homocysteinilation of the glucocorticoid receptor.
Overview and general recommendation:
The manuscript by Arnaud Michel and colleagues examined the effects of vitamins B9 and B12 on brain development, behavior and the involvement of glucocorticoid receptors in these processes in the ARC and VMN of the hypothalamus.
The entire paper is written in such a way that it is very difficult to understand. The authors should enlist the help of a native English speaker and rewrite the paper in more academic English. Using grammar software, the revised manuscript jumped from 90% to 97% of clarity and writing quality.
I don't recommend it for publication as is. The manuscript needs to address several important issues, as outlined in my review, before it can be considered further.
Here are some comments for improvement.
• The title: “Folate and vitamin B12 gestational deficiency disturb glucocorticoid response
in hypothalamus through N-homocysteinilation of the glucocorticoid receptor” – I suggest using either “folate and cobalamin” or “Vitamins B9 and B12”.
Suggestion for the title: Folate and Cobalamin Deficiencies During Pregnancy Disrupt the
Glucocorticoid Response in the Hypothalamus through N-homocysteinilation of the Glucocorticoid Receptor.
You don’t need a period in the title. Authors agree the suggestion
Abstract:
• Lane 18: The sentence “Besides neural tube defects in case of major consequences, several mild deregulations also lead to deleterious effect after birth, and folate supplementation recommended in some countries usually stop at the end of the first trimester.” is poorly written and difficult to understand. Did the authors mean to say the following? “In many countries, folate supplementation is discontinued after the first trimester, although adverse effects may occur after birth because of some mild misregulations, with neural tube defects being the most severe. Yes, the sentence is confusing. We propose a new version including the suggestion.
- Lane 20: “in this context” suggestion “under these conditions” Modified
- Lanes 22 and 26: “post-traductional”??? Did the authors mean “epigenetic regulation and
post-translational modifications” Of course yes. Sorry. This is modified in the whole text.
- Lane 24: “Our data showed that a deficiency in folate and vitamin B12 during” – it is said
“Our data showed that a deficiency of folate and vitamin B12 during...” Modified
- Lane 24: it should be said “GR signaling” not “GR signalization” Modified
- Lane 26: double “n” in “inn” – the paper needs serious proofing and spelling checking Corrected
- Lane 26: “novel” instead of “new” Modified
- Lane 28: “GR pathway” it should be said “GR signaling pathway” Modified
Material and methods:
• I suggest inserting a diagram of experimental design in this section to make it more understandable. A proposition is added with figures
- Also, the explanation of hypothalamic preparation for immunohistochemistry should be added in the same section. It is not clear whether the authors performed perfusion of the animals, and if not, it should be stated how they avoided the additional fluorescence due to the presence of blood. Precisions are given at the end of the section "Animals and tissues collection". In addition, some errors in the procedure have been corrected.
- Lane 143: It is said “Table 2” in the text, and the table is marked as “1” lane 151. Corrected
- Lanes 172-174, it is said “Days of testing (D11, D14, and D18) were chosen after postnatal day 10, where the spontaneous exploration of rat offspring is significant.” When it is written in this way, one could assume that the testing was performed in D11, D14 and D18 staring from tenth day. I suggest to rephrase it this way: “As part of a global neurofunctional approach, various behaviors such as locomotion, exploration, and olfaction were studied using a corridor apparatus. Because spontaneous exploration of rat offspring is significant after postnatal day 10, days 11, 14, and 18 (D11, D14, and D18) were chosen.” Yes, thanks. Modified.
Results section:
- Some subtitles in this section are written in upper case, others in lower case. This should
be balanced.
- All graphs should be drawn again with the control value set to 1 or 100, which would
help to better evaluate the treatment-related changes and facilitate the interpretation of the
results obtained. Authors prefer the present version of graphs which allow to appreciate real levels of means and standard deviations. Nevertheless, relative comparisons between experimental groups and controls are given in the text of result section.
- Lane 194: It is not necessary to always use both the full name and the abbreviation in the
text. It should be done only once, at the first mention, and that is quite sufficient. Please
correct this. Corrected
- Lane 199: In the text it says “Table 1”, and the table is marked as “2” Lane 203 Corrected
- Lane 203: “vitamineB12” you need a space between vitamin and B12, and you do not
need an “e” in “vitamine” Corrected
- Lanes 209-211: It says “Investigating the expression level of GR, our results show that in the hypothalamic cell line and hypothalamus brain tissue, a vitamin B9 deficiency leads to significantly reduced mRNA and protein of this receptor compared with controls (figure 1, 2A).” What cell lines are we talking about? You haven't mentioned them anywhere in the text before. Explain the role and the type of cell lines in this experiment? Sorry, the paragraph was accidentally deleted just before submission.
- Also, Lane 209: “Investigating the expression level of GR” you need to change it to
“Investigating the expression of protein and mRNA levels of GR” Modified
- The whole paragraph between lines 209 and 219 should be rewritten. We hope that it is now acceptable
- Lane 223: “Figure 1” a space is missing Corrected
You cannot present data that you haven’t mentioned in the Materials and Methods or elsewhere
in the paper. As mentioned above, the paragraph was accidentally deleted during our internal revisions. The protocol concerning cell culture is now presented in the method section.
- Lane 225: It is said “Data are means ±SD from 3≤n≤8 samples” – This is not how results
can be represented when working with cell lines. What is this supposed to represent? It
seems like a copy-paste analysis of animal experiments. Yes, it is modified
- Lane 241: “a higher magnification”, the magnification should be specified. Corrected
- Lane 269: The results for the cell lines are presented, without earlier mentioning of these
cells. Now presented in the method section
- Lane 295: The section “Behavioral characterization of rat offspring” text is not referring
to the right Figure. Corrected
- There are typographical and grammatical errors throughout the text and they should be
carefully corrected. Using a grammar software, the revised text achieved 97% of clarity and writing quality.

Reviewer 2 Report
This manuscript is written poorly and has not been edited/revised before submission. The “material and methods” section is copied word-to-word from a previous 2017 manuscript (ref # 26 in this manuscript- https://www.ncbi.nlm.nih.gov/pmc/articles/PMC5533871/). Method section is missing content of what and how experiments were done. For example, cell line data is mentioned directly in the results, without any mention of them in the methods. The table number in the text does not match the Table number for data or methods.
The supplementation of folic acid is done in rat dams but they keep saying supplementation of B9 in offspring/rat pups.
There are so many typos in this manuscript.
References are mentioned in the body but not in ref section.
There are so many typos in this manuscript.
The choice of words change the meaning of the experimental design completely. For example, the supplementation of folic acid is done in rat dams but they keep saying supplementation of B9 in offspring/rat pups.
Author Response
Authors thank the reviewers for their help and advices improving the manuscript.
We apologize if the quality of English writing appears not satisfying. We use a software to improve our first composition for grammar and clarity. The revised manuscript has obtained 97% of clarity and quality of writing. In the revised version, all modifications appear highlighted for rapid comparison (pink for reviewer1 comments, green for reviewer2, and light blue for reviewer3).
Please find below, our responses to each reviewer's comment.

Reviewer 3 Report
Michel et al. confirm that nutritional status of dams is essential to neurodevelopment of offspring. Overall, the work is well designed and executed. Although, I have same concerns:
1 - Does giving condensed milk to pregnant women increase their weight? Having condensed milk high levels of sugar could not have adverse effects? In my opinion, it would be relevant to have a control without condensed milk and compare glycemic parameters and weight.
2 - The authors only performed behavioral tests in early-life. which could hide behavioral alteration later in life. I suggest to add to this study behavior studies for anxiety and cognition (at least) in juvenile and/or adulthood.
Author Response

(The authors gave the same response as above.)

Round 2
Reviewer 2 Report
The authors have successfully revised all the concerns.
Reviewer 3 Report
I am satisfy with author's reply.